# Uricase-Deficient Larval Zebrafish with Elevated Urate Levels Demonstrate Suppressed Acute Inflammatory Response to Monosodium Urate Crystals and Prolonged Crystal Persistence

**DOI:** 10.3390/genes13122179

**Published:** 2022-11-22

**Authors:** Tanja Linnerz, Yih Jian Sung, Leah Rolland, Jonathan W. Astin, Nicola Dalbeth, Christopher J. Hall

**Affiliations:** 1Department of Molecular Medicine and Pathology, Faculty of Medical and Health Sciences, University of Auckland, Auckland 1023, New Zealand; 2School of Medicine, Faculty of Medical and Health Sciences, University of Auckland, Auckland 1023, New Zealand

**Keywords:** gout, zebrafish, uricase, urate, hyperuricemia, inflammation, MSU crystals, CRISPR/Cas9

## Abstract

Gout is caused by elevated serum urate leading to the deposition of monosodium urate (MSU) crystals that can trigger episodes of acute inflammation. Humans are sensitive to developing gout because they lack a functional urate-metabolizing enzyme called uricase/urate oxidase (encoded by the *UOX* gene). A hallmark of long-standing disease is tophaceous gout, characterized by the formation of tissue-damaging granuloma-like structures (‘tophi’) composed of densely packed MSU crystals and immune cells. Little is known about how tophi form, largely due to the lack of suitable animal models in which the host response to MSU crystals can be studied in vivo long-term. We have previously described a larval zebrafish model of acute gouty inflammation where the host response to microinjected MSU crystals can be live imaged within an intact animal. Although useful for modeling acute inflammation, crystals are rapidly cleared following a robust innate immune response, precluding analysis at later stages. Here we describe a zebrafish *uox* null mutant that possesses elevated urate levels at larval stages. Uricase-deficient ‘hyperuricemic’ larvae exhibit a suppressed acute inflammatory response to MSU crystals and prolonged in vivo crystal persistence. Imaging of crystals at later stages reveals that they form granuloma-like structures dominated by macrophages. We believe that *uox*^−/−^ larvae will provide a useful tool to explore the transition from acute gouty inflammation to tophus formation, one of the remaining mysteries of gout pathogenesis.

## 1. Introduction

Gout is the most common form of inflammatory arthritis, affecting 41.2 million people worldwide in 2017 [1]. Despite the high prevalence of gout, the condition is incompletely understood and treated. Gout is caused by the deposition of monosodium urate (MSU) crystals in and around joints. MSU crystals trigger episodes of extremely painful acute inflammation (called ‘gout flares’) characterized by the infiltration of inflammatory innate immune cells (neutrophils and macrophages) that typically resolves within 1–2 weeks [2]. Repeated acute gout attacks can develop into a chronic form of the disease over time, characterized by the formation of painful and debilitating granuloma-like structures, called tophi [3]. Tophaceous gout leads to cartilage, bone and tendon damage, resulting in poor health-related quality of life. Furthermore, tophi can ulcerate and become infected, leading to bone and joint infection [4,5,6,7], and their presence is an important risk factor for mortality in people with gout [8,9]. Granuloma-like tophi are characterized by densely packed MSU crystals surrounded by inflammatory cells [10]. How the early formation of tophi initiates and the underlying mechanisms controlling the progression to tophaceous gout remain largely unknown.

While the occurrence of gout relies on many factors, including genetic predisposition and environmental factors, chronic hyperuricemia greatly enhances risk [11,12]. Hyperuricemia develops when urate levels in the blood are elevated and reach a saturation threshold above 6.8 mg/dL. Urate is a common waste product from purine metabolism. The major sources of purines in the body derive from cellular energy molecules such as ATP, integral components of DNA and RNA, or are ingested through purine-rich foods (e.g., meats, seafood, shellfish and alcohol). Most organisms can degrade urate into allantoin utilizing the enzyme uricase or urate oxidase (encoded by the *UOX* gene), transforming a poorly water-soluble molecule into a water-soluble state, which facilitates excretion from the body [13,14].

During evolution, the accumulation of nonsense and missense mutations in the *UOX* gene in higher apes and humans has led to the loss of a functional uricase enzyme [15], resulting in their sensitivity to develop elevated serum urate levels and the accompanying consequences, such as gout. Due to the presence of a functional uricase enzyme, most studies investigating gout in rodent models rely on chemical or diet-based induction of hyperuricemia [16,17] or the generation of uricase mutants. To date, two *Uox* knockout (KO) mouse models and a radiation-induced mouse line lacking the *Uox* locus have been described that possess overlapping phenotypes, including hyperuricemia and renal comorbidities [18,19,20]. An obstacle to using these uricase-deficient mice is their poor survival. Of note, A CRISPR/Cas9-generated uricase-deficient rat has also been generated that appears to have a more limited impact on survival [21,22].

Optically transparent larval zebrafish (*Danio rerio*) provide an ideal platform to investigate inflammatory processes in the context of an intact microenvironment [23]. By 2 days post fertilization (dpf), larval zebrafish are populated with both neutrophils and macrophages that are functionally equivalent to those in mammals [24,25,26,27]. Furthermore, the human and zebrafish genomes are highly conserved, facilitating the translation of mechanistic findings and disease modelling into the human disease context. We previously established a larval zebrafish model of acute gouty inflammation by injecting MSU crystals into the hindbrain ventricle of 2 dpf larvae [28]. Through live imaging the behaviour of macrophages and neutrophils in response to MSU crystals, for the first time within an intact animal model, we uncovered a previously unrecognized molecular mechanism of macrophage activation [28]. Confirming the translational impact of this discovery, we also showed that pharmacologic targeting of this new pathway alleviated inflammation in rodent and human models of acute gout [28]. Although useful for studying the acute inflammatory response to MSU crystals, our larval zebrafish model does not accurately replicate the elevated levels of urate that is often superimposed on the inflammatory response to MSU crystals in humans. Furthermore, when injected into wild-type (WT) larvae, MSU crystals are rapidly cleared following the acute inflammatory response, preventing longer-term investigations into the host response.

To help address these limitations, we generated the first uricase-deficient zebrafish model using CRISPR/Cas9 gene editing to investigate the impact of urate levels on the larval response to MSU crystals. We show that *uox* KO (*uox*^−/−^) larvae have significantly reduced expression of *uox* in the liver and elevated urate levels at early larval stages. In contrast to existing mouse models, we show that *uox*^−/−^ zebrafish remain healthy, show no signs of embryonic mortality and are fertile when compared with WT zebrafish. While *uox*^−/−^ larvae maintain typical numbers of innate immune cells (neutrophils and macrophages), their acute inflammatory response towards MSU crystals, while resolving with similar kinetics to that of WT larvae, is significantly reduced in magnitude. Coincident with this suppressed inflammatory response, injected MSU crystals persist at least until 4 days post injection (6 dpf), a phenotype that can be rescued by injecting *uox* mRNA. Observing MSU crystals at these later stages revealed that they begin to aggregate into macrophage-dominated granuloma-like structures. We believe that our uricase-deficient (‘hyperuricemic’) zebrafish gout model, which is analogous to the human condition in terms of uricase, will provide a valuable tool to study the host response to MSU crystals following the acute inflammatory response and may shed light on the mechanisms controlling the transition from acute gouty inflammation to tophus formation.

## 2. Materials and Methods

### 2.1. Zebrafish Husbandry and Fish Lines

Zebrafish were maintained and bred with the approval of the University of Auckland Animal Ethics Committee (approval number AEC22563). Zebrafish (*D. rerio*) embryos were obtained from natural group spawnings and raised at 28 °C in E3 medium supplemented with 0.003% phenylthiourea (PTU) to inhibit pigmentation when used for live imaging. WT AB zebrafish were obtained from the Zebrafish International Resource Center (ZIRC). The following transgenic lines were used in this study: *Tg(lyz:EGFP)*^nz115^ [24], *Tg*(-8.*mpx:*KalTA4)^gl28^;*Tg(UAS-E1b:nsfB-mCherry)*^c264^ [29] herein referred to as *Tg(mpx:NTRmCherry)*, *Tg(mfap4:EGFP)*^nz54^ and *Tg(mfap4:mCherry)*^nz55^ lines to label neutrophils and macrophages.

### 2.2. Phylogenetic Analysis and Multiple Sequence Alignment (MSA)

Uricase protein sequences of Drosophila melanogaster (FBgn0003961), denticle herring (ENSDCDG00000029202), Atlantic salmon (ENSSSAG00000063116), rainbow trout (ENSOMYG00000005477), zebrafish (ENSDARG00000007024), golden-line barbel (ENSSGRG00000022452), common carp (ENSCCRG00000072036), xenopus tropicalis (ENSXETG00000008648), capuchin (ENSCCAG00000021796), macaque (ENSMMUG00000000404), gibbon (ENSNLEG00000028021), human (ENSG00000240520), bonobo (ENSPPAG00000042063), chimpanzee (ENSPTRG00000043173), gorilla (ENSGGOG00000042098), dog (ENSCAFG00000020315), pig (ENSSSCG00045036383), mouse (ENSMUSG00000028186), rat (ENSRNOG00000016339) and cow/cattle (ENSBTAG00000024255) were retrieved from ensembl.org. For phylogenetic analysis and MSA, BLASTP 2.13.0+ was used. The phylogenetic tree was generated using the neighbour joining method and the Grishin distance model with a maximum sequence difference of 0.85. To generate the graphic model of amino acid conservation between organisms, the MSA viewer version 1.22.0 was utilized with a conservation colour coding scheme.

### 2.3. Generation of Uox Mutants and Genotyping

For generating the *Tg(mfap4:EGFP)*^nz54^ and *Tg(mfap4:mCherry)*^nz55^ lines, a 1.6 kb *mfap4* promoter construct (*mfap4*-p5E, a generous gift from David Tobin [30]) was used to generate EGFP and mCherry reporter constructs using the gateway compatible Tol2kit [31]. For transgenesis, 1-cell stage embryos were injected with 30 pg of the final Tol2 vector, along with 30 pg Tol2 mRNA. For CRISPR/Cas9-mediated generation of the *uox* mutant, a single-guide RNA target site was designed using the online tool CHOPCHOP. The guide RNA was generated by annealing two overlapping oligonucleotides at 95 °C for 5 min, followed by 45 min at 22 °C. The samples were diluted 20-fold and ligated into the *Bsa*I-linearized pDR274 plasmid (kind gift from Keith Joung; Addgene plasmid 42250). The guide RNA was generated using the MEGAshortscript T7 kit (Ambion) and 500 pg recombinant Cas9 protein (PNAbio) and 250 pg of the guide RNA were co-injected per embryo at the 1-cell stage. The guide RNA sequence for *uox* was: 5′-GGTGATGGTTCCCTTCACGCCGG-3’ (PAM sequence underlined). Embryos and potential founder adults were screened using primers that flanked the target region (forward: 5′- TCAAATCAGGTAAAGGAAGGGA-3´ and reverse: 5´-TACCCCTTTTAGTTT-GGCAAGA-3´) and successful mutagenesis was confirmed using the T7 Endonuclease I assay. In brief, genomic DNA was extracted from pools of 24 hpf stage embryos or tail fin clips using the HotSHOT method [32] and PCR was performed with the above primer set. Amplicons were then denatured, reannealed and digested with T7 Endonuclease I, as previously described [33]. Individual mutant alleles were identified by Sanger sequencing of amplicons.

### 2.4. Urate Quantification

To measure urate levels within larvae, exactly 40 larvae per time-point and genotype were thoroughly dissociated in 125 μL ice-cold sterile PBS using a microfuge pestle, in biological triplicate. Homogenized larvae were then centrifuged for 10 min at 15,000× *g* at 4 °C. 100 μL of supernatant was then collected for urate measurements using a Cobas c311 autoanalyzer (Roche Diagnostics, Basel, Switzerland) and the UA2 pack for the quantification of uric acid in human serum, plasma or urine. Urate amounts were then normalized to the number of larvae and displayed at pmol/larva.

### 2.5. MSU Crystal Synthesis and Microinjection into Zebrafish Larvae

Endotoxin-free MSU crystals were prepared from uric acid (Sigma-Aldrich, St. Louis, MO, USA) as previously described [34]. For microinjection, MSU crystals were prepared as previously described [28], with the following modifications. In brief, 15 mg MSU crystals were dissociated by repeated passage through 18- and 22-gauge needles (Terumo, Elkton, MD, USA) in filter-sterile HBSS. Subsequent sonication generated crystals of approximately 2 μm in length. Aliquots of dissociated MSU crystals in HBSS were stored at −20 °C. MSU crystals or HBSS alone (2 nL) were injected into the hindbrain ventricle of 2 dpf anesthetized zebrafish larvae immobilized in 3% methylcellulose. The +7 bp insertion mutation generated here in the *uox* gene has been given the following allele designation *uox*^nz7^.

### 2.6. RNA Extraction and Quantitative PCR (qPCR)

RNA was extracted from groups of 10 4 dpf larvae using the RNeasy Plus Mini Kit (Qiagen, Hilden, Germany) according to the manufacturer protocol and reverse transcribed into cDNA using the iScript Kit (BioRad, Hercules, CA, USA). qPCR was performed using the iTaq Universal SYBR Green Supermix (BioRad) and the QuantStudio 6K Flex Real-Time PCR System (Life Technologies, Thermo Fisher Scientific, Waltham, MA, USA). The following primer pairs were used for *uox:* exon 1-2, forward 5´-CCTCAAATCAGAATGTGGAGTTC-3´, reverse 5´-ATGGTTCCCTTCACGCCG-3´; exon 2-3, forward 5´-CACCGTCCACGCTCTTGC-3´, reverse 5´-GACATGATTGAACGC-TGTGAGG-3´; exon 5-6, forward 5´-ACACAAACTGGATTTGAGGGC-3´, reverse 5´-TAACAGCTTTCCATGCAGCG-3´; exon 6-7, forward 5´-GGAAAGCTGTTAAGG-ACACCG-3´, reverse 5´-TCTCAATTT-CCTCAACCTCTGG-3´. The following primer pair were used for *ef1a*, forward 5´-GAG-AAGTTCGAGAAGGAAGC-3´, reverse 5´-CGTAGTATTTGCTGGTCTCG-3´. Each qPCR experiment was performed in biological and technical triplicates. Expression levels were normalized to *ef1α* and calculated using the ΔΔCt method.

### 2.7. Whole-Mount In Situ Hybridization (WMISH)

WMISH was performed on 4% paraformaldehyde-fixed larvae using digoxigenin-labelled (DIG-labelled) antisense riboprobes (Roche) with NBT/BCIP (Roche) colour precipitation as previously described [35]. RNA probes were generated by linearization of TOPO-TA vectors (Invitrogen) containing the PCR-amplified (500 bp) cDNA sequence (forward: 5′-CATGCTTTCATCCACTGCCC-3′ and reverse: 5′-AGGAAACATCACAGGCACTGT-3′). Embryos were imaged in 100% glycerol, using an AxioZoomV16 stereomicroscope (Zeiss, Oberkochen, Germany) equipped with an Axiocam 208 colour camera (Zeiss).

### 2.8. Flow Cytometry

Larvae were dissociated for flow cytometry as previously described [36]. In brief, groups of 20 2 dpf larvae were euthanized in 1x Ringer’s solution on ice for 15 min before deyolking. Embryos were dissociated in 1x PBS and 0.25% Trypsin at 28 °C for 45 min. The cell suspension was pelleted at 1500 rpm at 4 °C and then resuspended in 1x PBS/10% foetal bovine serum. The cell suspension was then filtered through a 22 µm cell strainer (Falcon) before performing flow cytometry on an LSR II flow cytometer (BD Biosciences, San Jose, CA, USA).

### 2.9. Immunostaining for Neutrophil Recruitment

Immunofluorescent labelling of neutrophils was performed as previously described [37]. In brief, between 12 and 15 WT and *uox*^−/−^ *Tg(lyz:EGFP)* larvae were fixed at the indicated timepoints post MSU crystal injection in 4% PFA. Neutrophils were immunostained using a chicken anti-GFP primary antibody 1:500 (Abcam, ab13970) and a goat anti-chicken Alexa Fluor 488-conjugated secondary antibody 1:500 (Invitrogen, Thermo Fisher Scientific; A11039). Stained larvae were kept for short term in PBS at 4 °C and mounted in 1% low-melting-point agarose in PBS for imaging on a Nikon C1 Eclipse confocal microscope. Images were analysed and quantified with FIJI (ImageJ).

### 2.10. Live Imaging

For live imaging, larvae were anesthetized with 125 μg/mL tricaine and mounted in 1% low-melting-point agarose in E3 media supplemented with tricaine and 0.003% PTU in a glass-bottom dish. For confocal imaging, larvae were imaged on a Zeiss LSM 710 inverted confocal microscope. The confocal reflection mode was used to enable quantification of the MSU crystal volume within hindbrains. For live epifluorescence imaging, larvae were imaged using an AxioZoomV16 stereomicroscope (Zeiss) equipped with an Axiocam 506 mono fluorescence camera (Zeiss). For live brightfield imaging, larvae were imaged using an AxioZoomV16 stereomicroscope (Zeiss) equipped with an Axiocam 208 colour camera (Zeiss). Images were analysed either with FIJI (ImageJ) or Volocity 3D image analysis software (PerkinElmer).

### 2.11. MSU Crystal Quantification Using Confocal Reflection Microscopy

MSU crystal-injected larvae were imaged on an inverted Nikon LSM710 confocal microscope using reflection microscopy throughout the whole hindbrain (approx. 45 µm in total) in Z-stacks of 3 µm steps. Reflection images of acquired Z-stacks were quantified using FIJI (ImageJ) software. Firstly, the Gaussian Blur filter with a sigma radius of 2.0 was applied to the images followed by the Maximum filter with a sigma radius of 1.0. Images were then converted to binary using the Otsu mask threshold parameters. Apparent and obscuring signals generated by overlying pigment cells were manually removed at this step to avoid false quantification of pigment-derived signals. MSU crystal volumes were measured using the 3D object counter plugin and the volume parameter. The sum of all 3D objects per hindbrain was individually calculated and plotted in GraphPad Prism 8.0.

### 2.12. Data and Statistical Analysis

All data are presented as the mean ± SD. Statistical significance of differences between 2 groups was determined using an unpaired, 2-tailed Student’s t-test or multiple unpaired Student’s t-tests with Holm–Sidak correction. The survival curve was analysed using the log-rank (Mantel–Cox) test. Statistical significance between more than 2 groups was determined using 2-way ANOVA and the Tukey’s multiple comparisons test. P values of less than 0.05 were considered statistically significant. All statistical analyses were performed using GraphPad Prism 8.0 (GraphPad Software).

## 3. Results

### 3.1. Zebrafish Uricase Is Highly Conserved

To investigate the degree of conservation of zebrafish uricase we performed a multiple sequence alignment of the amino acid (aa) sequences of uricase from selected organisms, ranging from *Drosophila melanogaster* to vertebrates, including mammals and hominids. This analysis revealed that the zebrafish uricase protein is highly conserved with other vertebrates, including that of the mouse (*Mus musculus*), chimpanzee (*Pan troglodytes*) and humans (*Homo sapiens*), with aa identities of 67%, 58% and 63%, respectively (Figure 1A). Phylogenetic analysis, based on full-length aa sequences, revealed that fish uricase orthologs cluster into a clade, distinct from those representing members of the primates and other mammals (Figure 1B). Given that hyperuricemia and resulting comorbidities, such as gout, only naturally develop in higher primates that lack a functional uricase protein (Figure 1B,C), modelling hyperuricemic conditions in animal models has proven to be challenging, requiring drug or genetic manipulations often in combination with a purine-rich diet. Furthermore, the inflammatory response to MSU crystals in humans often occurs in the presence of elevated serum urate, compared to animal models possessing a functional uricase protein. For these reasons we sought to generate a zebrafish *uox* KO line.

### 3.2. Generating a Zebrafish Uox Knockout Line

Expression analysis of *uox* in zebrafish using whole mount in situ hybridization (WMISH) showed expression at 2 dpf in the developing liver, with weaker diffuse expression in the head and along the horizontal myoseptum (Figure 2A). By 4 dpf, this expression pattern remained, with strong expression in the liver and weaker expression in extrahepatic tissues (Figure 2A), consistent with a previous study [38]. To generate the *uox* knockout line, we used CRISPR/Cas9 gene editing to target exon 2 and create a premature termination codon early in the transcript (Figure 2B). Potential founder fish (F0) were individually outcrossed to WT, and pools of F1 embryos were screened by PCR and the T7 endonuclease I assay (Figure 2C). Embryos from positive founder fish were raised to adulthood, and the specific mutations were identified by Sanger sequencing. A +7 bp insertion mutation was identified that was then bred to homozygosity, hereafter referred to as *uox*^−/−^ (Figure 2B,D). This mutation was predicted to cause a premature termination codon after the first 30 aa of the 303 aa uricase protein.

### 3.3. uox^−/−^ Larvae Possess Elevated Levels of Urate While Adults Are Viable, Healthy and Fertile

We next investigated how the +7 bp mutation in the *uox* gene influenced *uox* expression. WMISH staining revealed significant downregulation of *uox* expression in 2 dpf *uox*^−/−^ larvae compared to WT larvae (Figure 3A). Of note, residual *uox* expression was detected within the liver of 4 dpf *uox*^−/−^ larvae following prolonged WMISH staining, albeit greatly reduced when compared to WT larvae (Figure 3A). To confirm this result, we performed qPCR on whole larvae at 4 dpf using several exon-spanning primer pairs across the *uox* transcript (Figure 3B). This analysis showed an almost complete absence of the *uox* transcript within *uox*^−/−^ larvae compared to WT levels. These findings strongly suggest that the mutant *uox* transcript likely undergoes nonsense-mediated mRNA decay and that the +7 bp mutation results in a null allele. Zebrafish *uox*^−/−^ embryos were born with an expected mendelian ratio and showed no obvious developmental defects (Figure 3C). In addition, the lifespan of *uox*^−/−^ zebrafish was similar to that of age-matched *uox^+/−^* and WT fish (Figure 3D), suggesting that uricase deficiency has no severe impact on zebrafish health. Of note, *uox*^−/−^ adults also remained fertile and healthy on a regular high purine diet suggesting that no specialized diet is necessary for their maintenance (Figure 3E). Supporting a hyperuricemic phenotype in *uox*^−/−^ zebrafish, measuring whole larvae urate levels at 3, 4 and 5 dpf revealed a significant increase in urate levels in *uox*^−/−^ larvae when compared to WT larvae (Figure 3F). Collectively, these data strongly suggest that the *uox* + 7bp mutation creates a null allele and that *uox*^−/−^ zebrafish remain viable, healthy and fertile despite possessing elevated urate levels.

### 3.4. MSU Crystals Persist When Microinjected into uox^−/−^ Larvae and Can Be Quantified Using Reflection Confocal Microscopy

Our previous studies examining the host response to MSU crystals using WT larval revealed that MSU crystals are rapidly cleared following a robust acute inflammatory response [28,39,40]. Given that the host response to MSU crystals in humans occurs in the absence of a functional uricase protein, we wanted to observe the response to crystals in *uox*^−/−^ larvae. MSU crystals were injected into the hindbrain ventricles of *uox*^−/−^ and WT larvae at 2 dpf and subsequently live imaged (using confocal transmitted light) at 1, 2, 3 and 4 days post injection (dpi). Consistent with our previous observations, MSU crystals were almost entirely cleared within WT larvae by 3 dpi (Figure 4A). This was in contrast to *uox*^−/−^ larvae, where MSU crystals persisted at least until 4 dpi (Figure 4A).

While transmitted light imaging allowed for qualitative analysis of MSU crystal burden, we wanted to develop a reliable, robust and accurate method for quantification of MSU crystal volumes within live larvae. Combining the optical transparency of larval zebrafish with the light reflective nature of MSU crystals, we developed a 3D measurement protocol to quantify the reflected signal from MSU crystals throughout the hindbrain ventricle after acquiring Z stacks by confocal microscopy. First, a reflection image was generated, pseudo-coloured in cyan, and manually inspected to ensure it closely mirrored the black crystal appearance in the brightfield image (compare Figure 4A (WT 1dpi panel) with B). Applying a Gaussian blur and Maximum filter before using an Otsu threshold mask resulted in a binary image that best recapitulated the original reflected signal without adding significant background noise (Figure 4B). Of note, at later stages of development (5 and 6 dpf), the larval skin becomes thicker and the first scales possessing pigments develop. Like MSU crystals, we found that these scale pigments also reflected light necessitating the manual removal of this pigment signal from individual Z stacks. The entire Z stack was then quantified using FIJI’s 3D object counter plugin and volume parameter. We used this quantification protocol to quantify MSU crystal burdens within individual *uox*^−/−^ and WT larvae imaged on consecutive days from 1 to 4 dpi. Similar to the transmitted light imaging (Figure 4A), this analysis showed that MSU crystals were almost completely cleared in WT larvae by 3 dpi (Figure 4C). This was in contrast to *uox*^−/−^ larvae that maintained significantly higher MSU crystal burdens, at least until 4 dpi (Figure 4C). To confirm that the observed MSU crystal retention within *uox*^−/−^ larvae resulted from uricase deficiency, we co-injected WT *uox* mRNA into *uox*^−/−^ and WT larvae and similarly quantified MSU crystal persistence. While injection of *uox* mRNA did not significantly enhance MSU crystal clearance within WT larvae, it restored the ability of *uox*^−/−^ larvae to clear MSU crystals at a rate comparable to that of WT larvae (Figure 4D). Taken together, these results show that MSU crystals persist within *uox*^−/−^ larvae suggesting that uricase and/or urate levels influence the host response to MSU crystals.

### 3.5. uox^−/−^ Larvae Maintain Normal Leucocyte Abundance but Show Dampened Neutrophil Recruitment to MSU Crystal Challenge

A clinical hallmark of acute gouty inflammation is the influx of neutrophils to the site of MSU crystal deposition [2,41]. We have previously shown that MSU crystals injected into larval zebrafish demonstrate a conserved robust innate immune cell response characterized by the activation of resident macrophages that direct the recruitment of neutrophils [28]. In WT larvae, we have shown that this MSU crystal-driven neutrophilic inflammation peaks at 6 hpi [28]. We next wanted to investigate the acute inflammatory response to MSU crystals within *uox*^−/−^ larvae that possess elevated levels of urate.

To ensure that any observed impact on the innate immune response to MSU crystals in uricase-deficient larvae was not the result of decreased steady-state neutrophil or macrophage numbers, we first assessed neutrophil and macrophage development in *uox*^−/−^ larvae through live imaging and flow cytometry quantification. Live imaging of neutrophils and macrophages within *Tg(lyz:EGFP)*;*uox*^−/−^ and *Tg(mfap4:EGFP)*;*uox*^−/−^ larvae, respectively, showed no apparent changes in the distribution or numbers of cells at steady-state when compared to WT larvae (Figure 5A,C). Similarly, flow cytometry quantification of neutrophils and macrophages from whole *Tg(lyz:EGFP)*;*uox*^−/−^ and *Tg(mfap4:EGFP)*;*uox*^−/−^ larvae revealed similar abundance to WT larvae (Figure 5B,D). Consistent with our previous study, when assessing the recruitment of neutrophils within WT *Tg(lyz:EGFP)* larvae, peak numbers were observed at 6 hpi (Figure 5E,F) [28]. This was in contrast to *uox*^−/−^ larvae that demonstrated significantly suppressed neutrophil recruitment, albeit with a similar temporal profile with that of WT larvae (Figure 5E,F). Collectively, these results show that despite having normal numbers of neutrophils and macrophages, the acute inflammatory response towards MSU crystals, as assessed by neutrophil recruitment, is dampened within *uox*^−/−^ larvae.

### 3.6. Persisting MSU Crystal Aggregates in uox^−/−^ Larvae form Structures That Are Dominated by Macrophages

In humans, MSU crystals persist for prolonged periods and trigger recurrent self-resolving episodes of acute inflammation. In patients with long-standing disease, some individuals also develop tophaceous gout with clusters of MSU crystals [42]. Here, we took advantage of the prolonged persistence of MSU crystals within *uox*^−/−^ larvae to live image the host’s innate immune response to MSU crystals, following the acute inflammatory response. Live imaging of macrophages within *Tg(mfap4:mCherry)*;*uox*^−/−^ larvae at 4 dpi revealed that extracellular MSU crystals aggregated into clusters that were accompanied by large numbers of macrophages, many of which contained phagocytosed crystals (Figure 6A). When viewed within *Tg(mfap4:EGFP)*;*(mpx:NTRmCherry)*;*uox*^−/−^ larvae that also possess red fluorescent neutrophils, neutrophils were present in lower numbers but in close association with MSU crystal aggregates (Figure 6B). Higher magnification analysis also revealed apparent neutrophil debris around the crystal aggregates (Figure 6B, inset). Collectively, this imaging analysis of the host response to MSU crystals following acute inflammation shows that extracellular MSU crystals cluster into aggregates composed of large numbers of crystal-laden and crystal-free macrophages with few numbers of neutrophils.

## 4. Discussion

Our previous work, modelling the acute inflammatory response to MSU crystals within WT larvae, provided a powerful platform to live image the innate immune cell response to MSU crystals, for the first time, within an intact animal model [28,39,40]. Although useful for studying acute gouty inflammation, a limitation of this model was the rapid clearance of MSU crystals, most likely the result of low endogenous urate levels, precluding longer-term analysis. Here we describe a CRISPR/Cas9-generated uricase-deficient ‘hyperuricemic’ zebrafish line in which microinjected MSU crystals persist for longer than in WT larvae, enabling the host response towards the crystals to be observed following the acute inflammatory response.

To our knowledge, the *uox*^−/−^ zebrafish mutant described here represents the first uricase-deficient zebrafish line and the first example of a zebrafish mutant with a genetically-induced ‘hyperuricemic’ phenotype. Given that hyperuricemia is used to describe increased *serum* urate concentration, here we use the term tentatively, when characterizing the elevated urate levels in whole *uox*^−/−^ larvae. We believe that the +7 bp indel carried by *uox*^−/−^ zebrafish represents a null allele given that expression of endogenous *uox* is almost completely abolished in *uox*^−/−^ larvae and because of elevated urate levels, a phenotype consistent with loss of uricase function. Furthermore, given that we can rescue the MSU crystal persistence phenotype observed within *uox*^−/−^ larvae back to WT levels by delivering exogenous *uox* mRNA, we believe that this phenotype is directly associated with uricase deficiency rather than off-target effects of CRISPR/Cas9 editing. We speculate that the persistence of microinjected MSU crystals within *uox*^−/−^ larvae is most likely the result of higher endogenous levels of urate preventing/delaying their dissolution. To date, three genetically induced mouse models of hyperuricemia have been described that affect the *Uox* gene [43]. Two independently generated *Uox* knockout lines share overlapping phenotypes, including hyperuricemia and nephropathology associated with renal crystal deposition [18,19]. Another radiation-induced null mutation in *Uox* (resulting in a paracentric inversion on chromosome 3 that includes *Uox*) also possesses hyperuricemia and has kidney defects [20]. All three of these uricase-deficient mouse lines suffer from poor postnatal survival, ranging from ~40 to 65% death by 5 weeks [18,19,20]. In contrast, *uox*^−/−^ zebrafish displayed comparable survival to WT fish, suggesting that uricase deficiency is better tolerated within zebrafish. Although we observe no obvious gross morphological defects within *uox*^−/−^ zebrafish, it will be interesting to perform a thorough histological examination of adult tissues, in particular the kidney and heart, given that chronic kidney and cardiovascular diseases are associated with hyperuricemia [43].

When measured at 3, 4 and 5 dpf, mean urate measurements in *uox*^−/−^ larvae were 1.7-, 2.4- and 2.2-fold greater than those of WT, respectively. Unfortunately, urate levels in 2 dpf larvae fell below the detectable limit in the urate measurement assay used in this study, preventing quantification at the time point. Although direct comparisons between larval zebrafish and adult mice are difficult to interpret, the increased urate levels in *uox*^−/−^ larvae are slightly less than the 3-to-10-fold increases observed in the serum of *Uox* knockout mice [18,19]. In future experiments, it will be interesting to measure urate levels in *uox*^−/−^ adults and investigate whether diet modulation can elevate urate to levels sufficient to promote endogenous MSU crystal formation.

Soluble urate has been demonstrated to promote inflammation, including that driven by innate immune cells, leading to the view that hyperuricemia may contribute to diseases such as hypertension and diabetes mellitus through stimulating inflammation [44]. In contrast, when examining the host response to MSU crystals within our ‘hyperuricemic’ *uox*^−/−^ larvae, we observed that the acute inflammatory response, as assessed by neutrophil recruitment, was significantly suppressed. Our result aligns with a recent study showing that hyperuricemia suppresses neutrophilic inflammation during sterile inflammation [45]. In a mouse air pouch model of MSU crystal-driven sterile inflammation, mice with hyperuricemia display significantly fewer neutrophils in pouch fluids compared to controls [45]. Through intravital microscopy, the authors discovered that soluble urate inhibits neutrophil adhesion and transmigration towards sterile stimuli due to intracellular changes in β_2_ integrin activity and recycling [45]. This negative impact of hyperuricemia on neutrophil function was a direct result of increased soluble urate uptake into neutrophils through urate transporters, resulting in changes to intracellular pH and cytoskeletal dynamics [45]. Soluble urate has also been shown to negatively regulate monocyte activation in MSU crystal-induced tissue inflammation [46]. Collectively, these results suggest that soluble urate can both promote and suppress inflammation. They also suggest that urate levels need to be considered when modelling the host response to MSU crystals and that hyperuricemic animal models (such as the *uox*^−/−^ zebrafish line described here) are more likely to better reflect the response to MSU crystals that occurs in humans. The suppressive effect of soluble urate on innate immune cell activity may also help explain why only a minority of patients with hyperuricemia develop symptomatic gout and why some hyperuricemic individuals with MSU crystal deposits fail to develop classical inflammatory symptoms [47].

We hypothesize that the suppressed acute inflammatory response towards MSU crystals observed within *uox*^−/−^ larvae directly results from elevated urate levels and its negative effect on innate immune cell function. In this study, we used neutrophil recruitment as a ‘hallmark’ readout of the acute inflammatory response. Given that macrophage activation is necessary for neutrophil recruitment during acute gouty inflammation [28,48], future studies will focus on examining macrophage responses towards MSU crystals in ‘hyperuricemic’ *uox*^−/−^ larvae. In addition, it will be interesting to investigate if the suppressed inflammatory response within *uox*^−/−^ larvae is restricted to triggers of sterile inflammation (like MSU crystals) or is a more general phenotype. To this end, studying neutrophil and macrophage responses towards infectious challenges will be informative.

In contrast to acute gouty inflammation, the cellular and molecular mechanisms underpinning tophus formation remain a mystery. MSU crystals are known to persist in gout patients where they trigger episodes of acute inflammation. This acute inflammation self-resolves largely through the production of macrophage-derived anti-inflammatory molecules and neutrophil extracellular traps (NETs) that can degrade pro-inflammatory mediators [2,42,49]. How deposited crystals transition from triggering these episodes of acute inflammation to forming a tophus is unclear. Previous work that provided the first detailed picture of the cellular architecture of the tophus suggests that the host immune response likely plays an important role in tophus formation and maintenance [10]. Quantitative immunohistochemistry of surgically obtained tophus samples revealed that the tophus has a complex and highly organized granulomatous structure divided into three main zones; an inner crystalline core of MSU crystals; a highly cellular coronal zone containing large numbers of macrophages (some of which have an epithelioid morphology or are multinucleated); and an outer fibrovascular zone that contains blood vessels and fibroblasts [10]. More recent studies have revealed that aggregated NETs are associated with MSU crystals within the tophus [49,50], suggesting that NETs may contribute to tophus formation by providing a physical ‘trap’ for MSU crystals and facilitating dense packaging [42,50].

Current approaches to study tophus formation are severely limited by difficulties in obtaining serial samples of human tophi for temporal analysis and the complete lack of in vivo experimental models of tophaceous gout. We hypothesize that the granuloma-like structures we observe in *uox*^−/−^ larvae following the acute inflammatory response, composed of extracellular MSU crystals, crystal-laden and crystal-free macrophages and neutrophils, may represent early stages in tophus formation. Furthermore, in future experiments we plan to investigate the source of neutrophil debris associated with these structures and determine if they are the result of NETosis. Future work will focus on developing imaging techniques to observe these structures in later stage *uox*^−/−^ larva (post 6 dpf, the latest timepoint analysed here) and in the presence of recently established NET-marking transgenic reporter lines [51].

## 5. Conclusions

Here we generated a zebrafish uricase-deficient zebrafish line (*uox*^−/−^) that shows significantly diminished expression of uricase in the liver and possesses elevated levels of urate. We show that microinjected MSU crystals persist within *uox*^−/−^ larvae, most likely the result of elevated endogenous urate levels, where they aggregate into macrophage-dominated structures resembling granulomas. We also show that *uox*^−/−^ larvae have a suppressed acute inflammatory response towards the crystals that we speculate is also driven by elevated urate levels. We believe that our uricase-deficient ‘hyperuricemic’ zebrafish model of gout, which is comparable to the human condition, will provide a tool to study both acute gouty inflammation and the early stages of tophus formation, a very poorly understood process in patients with long-standing disease.

## Figures and Tables

**Figure 1 genes-13-02179-f001:**
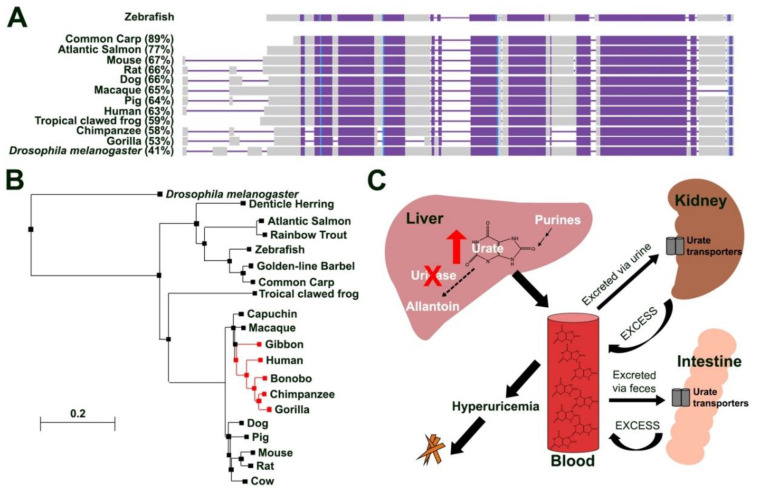
Zebrafish uricase is highly conserved. (**A**) Schematic illustrating amino acid conservation between zebrafish Uox with urate oxidases from other fish species (common carp (*Cyprinus carpio carpio*), Atlantic salmon (*Salmo salar*)), rodents (mouse (*Mus musculus*) and rat (*Rattus norvegicus*)), tropical clawed frog (*Xenopus tropicalis*), dog (*Canis lupus familiaris*), pig (*Sus scrofa*), primates (macaque (*Macaca mulatta*), gorilla (*Gorilla gorilla gorilla*), chimpanzee (*Pan troglodytes*), human (*Homo sapiens*)) and *Drosophila melanogaster* (percentages show amino acid conservation with zebrafish Uox). Alignment columns with no gaps are coloured in blue and violet. Violet colour indicates highly conserved regions (same residue in all alignment rows) and blue indicates less conserved regions. Unaligned residues are shown in grey. (**B**) Phylogenetic analysis of zebrafish Uox using BLASTP and the neighbour joining method with a Grishin distance model. Species lacking a functional uricase protein are highlighted in red. (**C**) Schematic illustrating urate metabolism in humans (that lack a functional urate oxidase (uricase) protein) and the contribution of hyperuricemia to MSU crystal formation. Purines are catabolized in the liver to urate, which is further degraded into allantoin in the presence of a functional uricase protein. Higher primates (including humans) lack a functional uricase making them sensitive to developing hyperuricemia, a risk factor for MSU crystal deposition and gout. Due to its water-insoluble properties, excess urate from the excretory pathway (kidney and intestines) is released back into the blood stream.

**Figure 2 genes-13-02179-f002:**
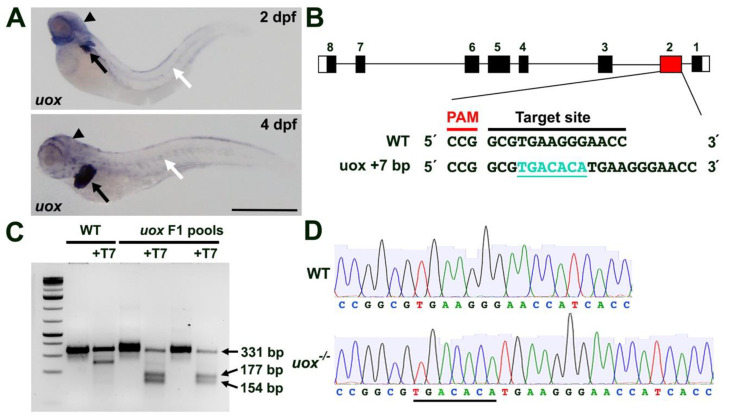
Generating a zebrafish *uox* mutant. (**A**) Expression of *uox* in 2 and 4 day post fertilization (dpf) larvae, as detected by WMISH (black arrows and arrowheads mark liver and head expression, respectively. White arrows mark horizontal myoseptum expression). (**B**) Schematic illustrating the intron (black line) and exon (black rectangles) structure of *uox* (with flanking UTRs as white rectangles). The position (red rectangle) of the CRISPR/Cas9 target site and the sequence are annotated, with the +7 indel mutation shown in green. (**C**) T7 endonuclease I assay (with and without T7 endonuclease) on pools of WT and F1 embryos. Successful editing is indicated by the cleavage products of 177 bp and 154 bp. (**D**) Sequencing chromatogram of amplicons from WT *uox* compared to a homozygous carrier of the +7 bp *uox* mutation (underlined). Scale bar: 500 µm.

**Figure 3 genes-13-02179-f003:**
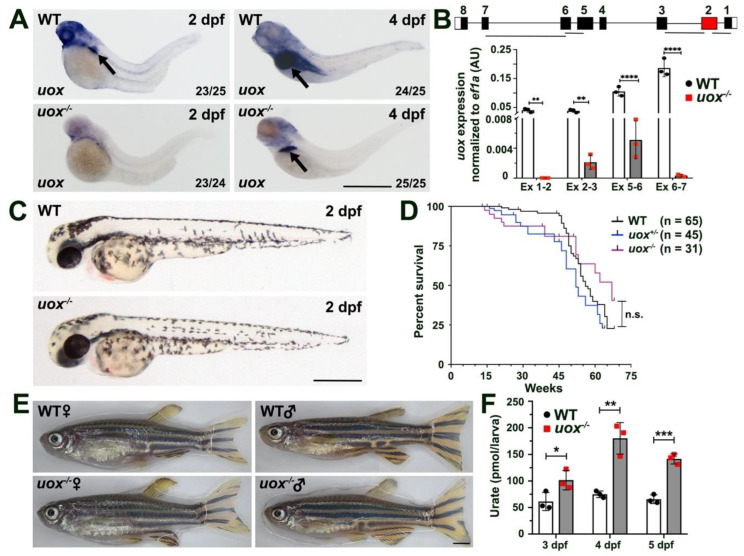
The +7 bp mutation in the *uox* gene results a null allele and *uox*^−/−^ zebrafish are morphologically normal, fertile and possess elevated levels of urate. (**A**) Expression of *uox* in 2 and 4 day post fertilization (dpf) WT and *uox*^−/−^ larvae, as detected by WMISH (arrows mark the liver). Numbers in the lower right corner of each panel indicate the frequency of larvae showing the displayed expression pattern. (**B**) Expression of *uox* in WT (white bars) and *uox*^−/−^ (grey bars) larvae at 4 dpf, as detected by qPCR using four independent exon-spanning primer pairs (Ex 1-2, Ex 2-3, Ex 5-6 and Ex 6-7, as shown in schematic). Data generated from three biological replicates, *n* = 10 larvae per group. Schematic shows the location of the four primer pairs in the *uox* gene. Statistical significance determined using multiple unpaired Student’s t-tests with Holm–Sidak correction. (**C**) Representative brightfield images of WT and *uox*^−/−^ larvae at 2 dpf. (**D**) Kaplan–Meier survival curve of WT, *uox*^+/−^ and *uox*^−/−^ fish from 1 month to 18 months of age. Statistical significance determined using the log-rank (Mantel–Cox) test. (**E**) Representative brightfield images of age-matched female and male WT and *uox*^−/−^ adult fish. (**F**) Quantification of urate (pmol urate/larvae) from dissociated 3, 4 and 5 dpf WT (white bars) and *uox*^−/−^ (grey bars) larvae, as detected using a Roche C311 autoanalyzer. Data generated from three biological replicates, *n* = 40 larvae per group. Statistical significance determined using multiple unpaired Student’s t-tests. Error bars in (**B**,**F**) represent means ± SDs. Abbreviations: n.s. not significant, * *p* < 0.05, ** *p* < 0.01, *** *p* < 0.001 and **** *p* < 0.0001. Scale bars: 500 µm in (**A**,**C**), 2 mm in (**E**).

**Figure 4 genes-13-02179-f004:**
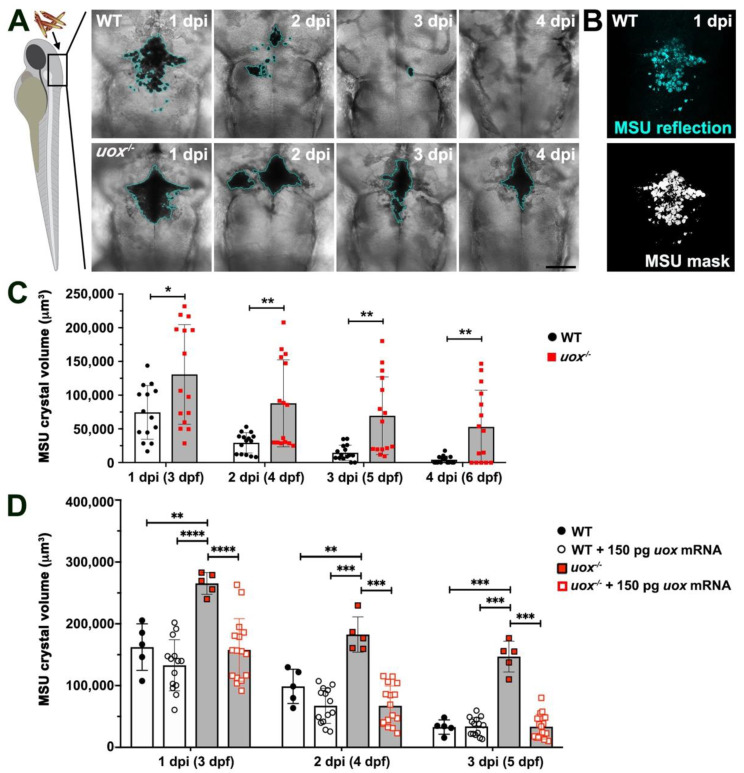
Microinjected MSU crystals persist in *uox*^−/−^ larvae. (**A**) Representative confocal imaging (transmitted light views, Z-projections) of WT and *uox*^−/−^ larvae at 1, 2, 3 and 4 days post-injection (dpi) with MSU crystals into the hindbrain ventricle (injected at 2 days post fertilization (dpf)). A cyan border outlines MSU crystals. (**B**) A representative reflection image of MSU crystals in a WT larva at 1 dpi (same larva as in (**A**)), pseudo-coloured in cyan. The lower panel shows a standard deviation Z–projection of the binary mask in black and white. (**C**) Quantification of MSU crystals (μm^3^), as detected in (**B**), within individual WT (white bars) and *uox*^−/−^ (grey bars) larvae at 1, 2, 3 and 4 dpi (3, 4, 5 and 6 dpf, *n* = 15 larvae per group). Statistical significance determined using multiple unpaired Student’s t-tests with Holm–Sidak correction. (**D**) Quantification of MSU crystals (μm^3^) within individual WT (white bars) and *uox*^−/−^ (grey bars) larvae at 1, 2 and 3 dpi (3, 4 and 5 dpf) and within WT (white bars) and *uox*^−/−^ (grey bars) larvae injected with 150 pg *uox* mRNA (*n*= 5 and 15 larvae per group for the non-mRNA-injected and mRNA-injected groups, respectively). Statistical significance determined using a 2-way ANOVA and Tukey’s multiple comparisons test. Error bars in (**C**,**D**) represent means ± SDs. * *p* < 0.05, ** *p* < 0.01, *** *p* < 0.001 and **** *p* < 0.0001. Scale bar 100 µm.

**Figure 5 genes-13-02179-f005:**
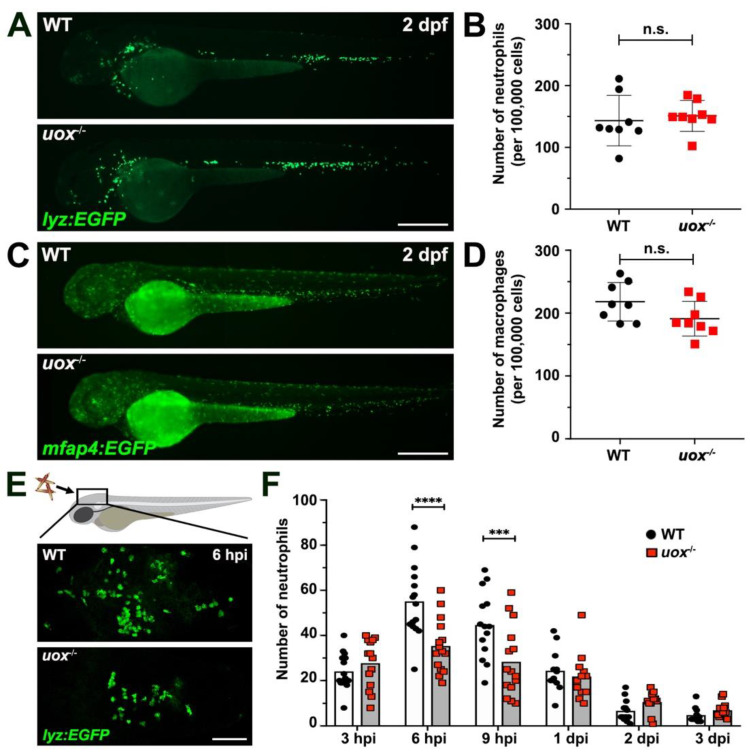
Uricase-deficient larvae demonstrate unaltered myelopoiesis and dampened neutrophil recruitment towards MSU crystal challenge. (**A**) Live imaging of neutrophils within 2 day post fertilization (dpf) WT and *uox*^−/−^ *Tg(lyz:EGFP)* larvae. (**B**) Flow cytometry quantification of neutrophils within 2 dpf WT and *uox*^−/−^ *Tg(lyz:EGFP)* larvae (*n* = 8 groups per genotype, 20 larvae per group). (**C**) Live imaging of macrophages within 2 dpf WT and *uox*^−/−^ *Tg(mfap4:EGFP)* larvae. (**D**) Flow cytometry quantification of macrophages within 2 dpf WT and *uox*^−/−^ *Tg(mfap4:EGFP)* larvae (*n* = 8 groups per genotype, 20 larvae per group). Statistical significance in (**B**,**D**) determined using unpaired two-tailed Student’s t tests. (**E**) Immunofluorescent detection of neutrophils within the hindbrain ventricle of WT and *uox*^−/−^ *Tg(lyz:EGFP)* larvae, at 6 h post MSU crystal injection (hpi). (**F**) Quantification of neutrophil recruitment, as detected in (**E**), within WT (white bars) and *uox*^−/−^ (grey bars) *Tg(lyz:EGFP)* larvae at 3, 6 and 9 hpi and 1, 2 and 3 days post MSU crystal injection (dpi), *n* = 12–15 larvae per treatment. Statistical significance determined using multiple unpaired Student’s t-tests with Holm–Sidak correction. Error bars in (**B**,**D**,**F**) represent means ± SDs. Abbreviations: n.s. not significant, *** *p* < 0.001 and **** *p* < 0.0001. Scale bars: 200 µm in (**A**,**C**,**E**).

**Figure 6 genes-13-02179-f006:**
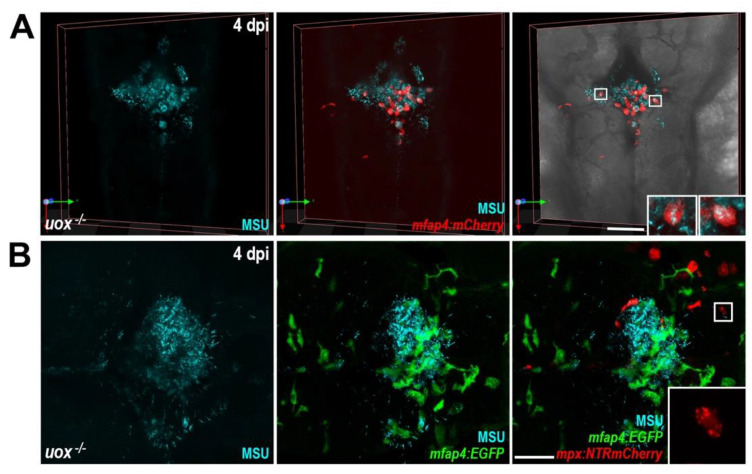
MSU crystal aggregates within uricase-deficient larvae are dominated by macrophages. (**A**) Live confocal imaging of MSU crystals within *Tg(mfap4:mCherry);uox*^−/−^ larvae at 4 days post injection (dpi). Insets, magnified views of boxed regions showing MSU crystal-laden macrophages. (**B**) Live confocal imaging of MSU crystals within *Tg(mfap4:EGFP);(mpx:NTRmCherry);uox*^−/−^ larvae at 4 dpi. Inset, magnified view of boxed region showing red fluorescent neutrophil debris. Scale bars: 100 µm in (**A**), 50 µm in (**B**).

## Data Availability

All raw data that contributed to the present study are available from the corresponding author C.J.H.

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
