# Peer review of "Uricase-Deficient Larval Zebrafish with Elevated Urate Levels Demonstrate Suppressed Acute Inflammatory Response to Monosodium Urate Crystals and Prolonged Crystal Persistence"

_genes, 2022, doi:10.3390/genes13122179_

Round 1

Reviewer 1 Report

I very much enjoyed this manuscript. The writing is excellent and the experimental work is well performed and well presented.

The work addresses an important gap in the animal models of gout, which the authors convincingly show has now been filled. The generation and validation of a uox mutant has been very convincingly shown, with persisting chronic inflammation following injection of MSU crystals.

Author Response

We thank Reviewer 1 for their positive comments about our manuscript. We conducted a thorough spelling and grammar check of the manuscript.

Reviewer 2 Report

Based on CRSPR/cas 9 knock out technique, Linnerz et al. report the generation of uricase-deficient zebrafish larvae that can be used as a disease model organism for studying human gout pathogenesis. This submission is well prepared and presents obvious scientific novelty. Hence, the reviewer recommends the acceptance of this submission. However,the authors may do a little more to promote the quality of their work. qRT-PCR shown in Fig.3B indicates a full suppression of uox expression in double knockout zebrafish, yet uox transcript is still clearly observed in fish liver (Fig.3A) when analyzed by WWISH. As the expression of uox transcript is detected by an antisense riboprobe, WWISH using a sense riboprobe should be included in Fig.3A  as a control.  Adding two photos obtained from sense riboprobe  hybridization, it would be easier for readers to know the real levels of uox transcripts in 2 and 4 dpf double knockout zebrafish. 

Author Response

When performing the WMISH expression analysis for 4dpf WT and uox-/- larvae, we extensively prolonged the staining time in an attempt to be transparent about residual expression in the uox mutant. As shown, there is low-level expression remaining, but it is still much weaker compared to WT expression. We now include the following text in the results section on page 8, lines 307-310: “WMISH staining revealed significant downregulation of uox expression in 2 dpf  uox-/- larvae compared to WT larvae (Figure 3A). Of note, residual uox expression was detected within the liver of 4 dpf uox-/- larvae following prolonged WMISH staining, albeit greatly reduced when compared to WT larvae (Figure 3A).” We have also modified the y axis in Figure 3B to highlight that we can also detect residual expression within uox-/- larvae by qPCR. This was previously not apparent in the y axis default format due to the high uox expression WT levels.